# Highly Sensitive and Selective Nanogap-Enhanced SERS Sensing Platform

**DOI:** 10.3390/nano9040619

**Published:** 2019-04-16

**Authors:** ChaeWon Mun, Vo Thi Nhat Linh, Jung-Dae Kwon, Ho Sang Jung, Dong-Ho Kim, Sung-Gyu Park

**Affiliations:** Advanced Nano-Surface Department (ANSD), Korea Institute of Materials Science (KIMS), Changwon 51508, Korea; apple1025@kims.re.kr (C.M.); vtnl94@kims.re.kr (V.T.N.L.); jdkwon@kims.re.kr (J.-D.K.); jhs0626@kims.re.kr (H.S.J.)

**Keywords:** hotspots, localized surface plasmon resonance, molecular filtration, surface-enhanced Raman spectroscopy, sensors

## Abstract

This paper reports a highly sensitive and selective surface-enhanced Raman spectroscopy (SERS) sensing platform. We used a simple fabrication method to generate plasmonic hotspots through a direct maskless plasma etching of a polymer surface and the surface tension-driven assembly of high aspect ratio Ag/polymer nanopillars. These collapsed plasmonic nanopillars produced an enhanced near-field interaction via coupled localized surface plasmon resonance. The high density of the small nanogaps yielded a high plasmonic detection performance, with an average SERS enhancement factor of 1.5 × 10^7^. More importantly, we demonstrated that the encapsulation of plasmonic nanostructures within nanofiltration membranes allowed the selective filtration of small molecules based on the degree of membrane swelling in organic solvents and molecular size. Nanofiltration membrane-encapsulated SERS substrates do not require pretreatments. Therefore, they provide a simple and fast detection of toxic molecules using portable Raman spectroscopy.

## 1. Introduction

Plasmonic nanogaps enable the detection of single molecule using surface-enhanced Raman spectroscopy (SERS) [1,2,3,4]. The near-field interactions at the plasmonic nanogaps create an enhanced electromagnetic (EM) field due to plasmonic coupling effects [5,6]. This enhancement of the EM field has been exploited in many plasmon-mediated applications, such as plasmonic photocatalysts [7,8], perfect optical absorbers [9,10], and plasmonic photovoltaics [11,12,13]. Therefore, numerous methods have been demonstrated for constructing highly sensitive SERS substrates. Among the many fabrication methods tested, the surface tension-induced leaning of metallic nanopillars offers a simple and desirable strategy for generating small nanogaps [14,15,16,17,18,19]. The surface tension-driven leaning method relies on attractive forces among high aspect ratio plasmonic nanopillars during solvent evaporation, inducing contact among two or more tall nanopillars. Target molecules may then be trapped among the collapsed nanopillars.

Meanwhile, biological cell membranes separate the interior of a cell from the external environment and permit the selective permeation of organic molecules and ions to control the flow of matter and to exchange energy between a cell and the environment [20,21]. Synthetic membranes filter heavy metal ions, pesticides, and other contaminants for water purification and food safety applications [22]. The main function of both natural and synthetic membranes is to allow for the selective permeation of target substances through the membrane pores. This functionality naturally suggests that semipermeable layers might be integrated into plasmonic nanostructures to provide high molecular selectivity. A variety of high-performance plasmonic nanostructures have been developed, including SERS substrates with large enhancement factors (EFs), low limits of detection (LODs), high signal uniformity, and large surface areas. However, the development of metal nanostructures with antibody-, enzyme-, or chemical-ligand-functionalized surfaces would require molecular selectivity. This type of surface treatment would involve capturing one specific target molecule through antibody–antigen or ligand–receptor binding interactions. A multiplex analysis could be achieved by preparing an arrayed substrate wherein each array region is subject to a different surface treatment to capture a specific target molecule.

Here, we report a new method for improving molecular selectivity by the molecular size exclusion effects of the nanofiltration membrane-encapsulated plasmonic nanostructures. We utilized a simple lithography-free two-step process (i.e., direct generation of polyethylene terephthalate (PET) nanopillars via maskless plasma etching and the subsequent deposition of Ag onto the PET nanopillars) to fabricate Ag nanoparticles (NPs) deposited onto high aspect ratio polymer nanopillars [17]. The plasmonic substrates were deposited with ultrahigh-density (53/µm^2^) top Ag NPs. The surface-tension-driven leaning effects applied to the Ag/PET nanopillars led to the assembly of high-density plasmonic nanogaps. These nanogaps provided a strong SERS effect with SERS EF of 1.5 × 10^7^. Enhanced molecular selectivity was obtained by the swelling properties of polydimethylsiloxane (PDMS) membrane in the organic solvent. The PDMS-encapsulating layer functioned as both a size-exclusion and a protective layer for plasmonic nanopillars. The molecular selectivity was demonstrated by observing that small methylene blue (MB) molecules (molecular weight (MW) of 319.85 g/mol) dissolved in chloroform diffused into the hotspots, whereas the large rhodamine 6G (MW of 479.02 g/mol) molecules were excluded. However, dissolving the MB molecules in deionized (DI) water resulted in the exclusion of even the small MB molecules because the PDMS membrane did not swell in DI water.

## 2. Materials and Methods

### 2.1. Fabrication of Plasmonic Nanostructures and PDMS Encapsulation

The PET plasma treatment was performed using a custom-built 13.56 MHz radio frequency (RF) ion etching instrument (SNTEK, Co. Ltd., Suwon, Korea). The inlet Ar flow rate and the reactor pressure were fixed, respectively, at 7.5 standard cubic centimeters per minute (sccm) and 55 mTorr during the pretreatment process. The plasma power was 100 W. Ag nanostructures were directly grown on PET nanopillars using a thermal evaporation system (SNTEK, Co. Ltd., Suwon, Korea) with a deposition rate of 2.2 Å/s. Mechanically stable plasmonic substrates were formed by depositing a ZnO thin film onto PET nanopillars at 100 °C using an atomic layer deposition system (Lucida M100, NCD Tech, Daejeon, Korea). Diethylene zinc (DEZ, Zn(C_2_H_5_)_2_) and DI water vapor were used as the zinc precursor and oxygen source, respectively [23,24]. The growth rate was 1.4 Å per cycle, and 60 cycles resulted in the deposition of an 8.4-nm ZnO layer onto the PET nanopillars. The PDMS used throughout the study was Sylgard 184 (Dow Corning Corp., Midland, MI, USA). The elastomer was mixed in a 10:1 ratio with its curing agent and then placed in a vacuum desiccator for 30 min to remove bubbles created during mixing. Samples were then prepared by pouring the PDMS solution into square Petri dishes, followed by curing at 90 °C overnight. The plasmonic substrates were placed on the pre-cured bottom 4-mm-thick PDMS layer, and 500 µL of a PDMS solution was poured over the 1 cm^2^ SERS substrates mounted on the bottom PDMS layer. The assembly was then cured at 90 °C for 2 h to yield a thin top PDMS layer.

### 2.2. Measurement and Chracterizations

The dark-field images were obtained using an optical microscope (Nikon, Eclipse 150, Tokyo, Japan) with a 150× objective lens (LU Plan Apo, NA 0.9, Toyko, Japan). The reflectance spectra of the plasmonic substrate were measured using spectrometers (Ocean Optics, USB4000, Dunedin, FL, USA) attached to an optical microscope (Nikon, Eclipse 150) with a 10× objective lens. The light reflected from the plasmonic substrate was collected through the same objective lens and analyzed by the spectrometer. The light reflected from the smooth silver mirror was measured as a reference. The Raman spectra were measured using a high-resolution dispersive Raman microscope (Horiba Jobin Yvon, LabRAM HR, Kyoto, Japan) with a 632.8 nm HeNe laser. A low laser power of 40 µW and a 50× objective lens were used and an accumulation time of 10 s was used for the Raman measurements. The surface morphologies were investigated using field emission scanning electron microscopy (FE-SEM; JSM-6700F, Jeol, Tokyo, Japan). The cross-sectional images of the plasmonic nanostructures were collected using transmission electron microscopy (TEM; JEM-2100 F, Jeol, Tokyo, Japan).

## 3. Results and Discussion

Figure 1a presents a schematic illustration of the nanolithography-free simple fabrication process of Ag/PET plasmonic nanostructures, consisting of the maskless plasma etching of smooth PET film and subsequent 100-nm-thick Ag deposition. Maskless plasma etching of polymeric materials has proven to be a reliable method for generating high-density self-organized polymer nanostructures [25,26,27,28]. The Ag NPs/PET nanopillars collapsed and leaned together through water-driven capillary leaning effects, and the leaning nanostructures were encapsulated by a PDMS membrane. PET has a low mechanical strength (Young’s modulus of 2.76 GPa) [29]; therefore, PET nanopillars with an aspect ratio of 3 tend to lean under the electron bombardment of the scanning electron microscopy (SEM, Appendix A) measurement. The surface-tension-driven leaning of the Ag NPs/PET nanopillars was observed by introducing a small water droplet onto the plasmonic substrates, which formed a contact angle of 72° (Appendix A). The region from which water had evaporated was darker than the region that had not contacted the water, as shown in Figure 1b. The outside and inside of the droplet were observed using a 150× objective lens with a high numerical aperture (NA) of 0.9. Figure 1c shows a dark field optical microscopy (OM) image of the capillary-driven leaning area. Interestingly, numerous vivid and different colors were observed, corresponding to wavelength-selective Rayleigh scattering under light illumination. Rayleigh scattering reflects the optical response (i.e., localized surface plasmon resonance (LSPR) wavelength) of plasmonic nanostructures in the far field [30,31,32]. As an LSPR is excited, strong Rayleigh scattering around the plasmonic nanostructures is generated and can be detected in the far field (Appendix A).

Figure 1d,e shows SEM images of the leaning Ag NPs (measured from within the water droplet region). The SEM images showed that tens of Ag NPs leaned together and clustered into a submicrometer area, and small nanogaps formed between the submicrometer collapsed regions. The density of Ag NPs was calculated to be 53/µm^2^—two or three times the densities obtained in previous studies [14,15,16,19,33]. The nanogap size was determined by collecting cross-sectional transmission electron microscopy (TEM) images of the leaning Ag NPs. The TEM images clearly demonstrated that large Ag NPs formed on top of the polymer nanopillars, and smaller Ag NPs formed on the sidewalls of the polymer nanopillars. The TEM images also showed that the collapsed Ag NPs formed one-nanometer-scale gaps with adjacent NPs (Figure 1f,g).

The thermal evaporation of Ag onto the PET nanopillars produced large Ag NPs on top of polymer nanopillars due to the directional deposition of Ag and the nonwetting 3D growth of the NPs onto the PET surface (Volmer–Weber growth mode) [16,34]. The mechanically unstable nanostructures tended to lean toward one another, even under small external forces, as shown in previous works [17]. The mechanical strength of the Ag nanostructures could be increased by depositing a thin layer of ZnO (8.5 nm thick) via atomic layer deposition (ALD) onto the PET nanopillars. A 100-nm-thick Ag layer was then thermally evaporated onto the ZnO/PET nanopillars. The ZnO thin film had a mechanical strength (Young’s modulus of 143 GPa) that was one order of magnitude higher than that of PET [35]; the thin layer deposition process enhanced the structural stability of the flexible PET nanopillars.

Figure 2a,b shows SEM images of the Ag nanostructures deposited onto the ZnO/PET nanopillars. Figure 2c shows a cross-sectional TEM image of the Ag nanostructures after applying DI water. The nanopillars stood upright, even after applying the capillary force exerted by the DI water (with a surface tension of 72.80 mN/m). Unlike the growth of Ag on the PET surface, Ag grew under a wetting mode on the ZnO surface [34], leading to Ag surface coverage on both the tops and sidewalls of the ZnO/PET nanopillars. The deposition of a hard inorganic thin layer increased the structural stability of the nanostructured surface.

Figure 3a shows the reflectance spectra of the non-leaning and leaning Ag nanopillars. The reflectance spectra measured from the leaning Ag nanopillars was lower than that in the non-leaning Ag/ZnO-deposited PET nanopillars over the entire visible wavelength range. The lower reflectance across the visible wavelength range corresponds to the enhanced near-field interactions among the nanogaps. Figure 3b presents a comparison of the SERS spectra of 15 µM MB-treated SERS substrates. A 5 µL MB aqueous solution was dropped onto both the non-leaning and leaning Ag nanopillar substrates and was allowed to dry for 1 h. The SERS signal enhancement measured from the leaning Ag NPs was 470% of the signal measured from the non-leaning Ag nanopillars under 632.8 nm excitation. The SERS EFs were the main parameter that determined the performance of the SERS substrate itself. Although the LOD has been reported to determine the performance of a SERS substrate, the LOD values depended on both the SERS substrate and the Raman instrumentation. Therefore, the average EFs are arguably a good measure of the structural specifications, rather than the single-molecule EFs, which represent the enhancement only at a specific site on the plasmonic nanostructures [36]. The calculated average EFs of the SERS substrates prepared with the leaning Ag NPs were 1.5 × 10^7^ (Appendix A). The average EF value exceeded the values reported previously because the density of Ag NPs surpassed the density prepared via maskless Si etching or nanoimprint lithography [14,15,16,19,33].

Selective permeation of target substances through cell membranes was achieved by the encapsulation of plasmonic nanostructures within nanofiltration membranes. Figure 4a shows the entrapment of MB molecules, depending on the degree to which the PDMS membrane swelled with the solvent. Because PDMS swells in the presence of chloroform, MB molecules dissolved in chloroform may diffuse into the swollen polymer matrix. However, MB dissolved in DI water may be excluded because PDMS does not swell in water. The thickness of the top PDMS membrane was 186 µm (Appendix A). It should be noted that the top layer should be very thin to reduce the diffusion length of the target molecules to the hotspot region. The PDMS-encapsulated SERS substrates turned bluish after a 10 min incubation period in a 1 mM MB chloroform solution (inset of Figure 4a). The SERS spectrum clearly shows that the MB SERS signal (black line in Figure 4a) was visible in the SERS substrates that were dipped for 10 min in the chloroform solution. However, the color of the PDMS-encapsulated SERS substrates that were dipped in the 1 mM MB aqueous solution did not change, even after a 20 h incubation period. In this case, only a PDMS background SERS signal was measured from the SERS substrates dipped for 20 h in the aqueous solution (Appendix A) [37]. The siloxane (Si-O-Si) backbone group in PDMS displays a very low Raman signal at 490 cm^−1^. The small Raman spectral peaks corresponding to C-Si-C moieties fell within the range 700 to 850 cm^−1^. The relatively large Raman shift at 1400 cm^−1^ corresponded to the CH_3_ vibrational band in PDMS (Appendix A). 

Finally, we investigated the molecular size-based filtration effects. We prepared a mixture of 0.1 mM MB and 0.1 mM R6G in a chloroform solution. The PDMS-encapsulated SERS substrates were then dipped into the solution. The SERS substrates were removed from the solution, rinsed with fresh chloroform several times, and dried under ambient conditions for 1 h. The inset in Figure 4b shows a photograph of the dried substrates. The transparent PDMS clearly turned bluish after dipping in the MB and R6G chloroform solutions. The origin of the color change was explored by collecting Raman measurements under 632.8 nm laser excitation. As shown in Figure 4b, the Raman bands characteristic of MB were clearly visible regardless of the dipping time, indicating that only MB molecules had diffused into the hotspot region. It should be noted that both MB and R6G are strongly Raman-active molecules. Therefore the R6G molecules may have been excluded by the PDMS membrane due to their large size and the small channels of the swollen polymer network. The R6G molecules may have additionally been physically adsorbed onto the PDMS top surface. The R6G molecules were finally rinsed away from the PDMS surface using chloroform.

## 4. Conclusions

In this article, we demonstrated size-exclusion-based molecular selectivity in the plasmonic detection platform. Ultrahigh-density plasmonic nanogaps were prepared by a direct maskless plasma treatment of a polymer surface in conjunction with the surface-tension-driven assembly of Ag nanopillars. These nanostructures produced an enhanced near-field interaction via coupled localized surface plasmon resonance among plasmonic nanoparticles. The high density of the small nanogaps yielded a high plasmonic detection performance, with an average SERS enhancement factor (EF) of 1.5 × 10^7^. The PDMS-encapsulating layer may function as both a size-exclusion and a protective layer for the general plasmonic nanostructures. The materials may potentially be encapsulated using stimuli-responsive hydrogels, such as pH- [38], temperature- [39,40], or moisture-sensitive hydrogels [41,42]. The encapsulation of plasmonic nanostructures may enable a variety of sensing applications.

## Figures and Tables

**Figure 1 nanomaterials-09-00619-f001:**
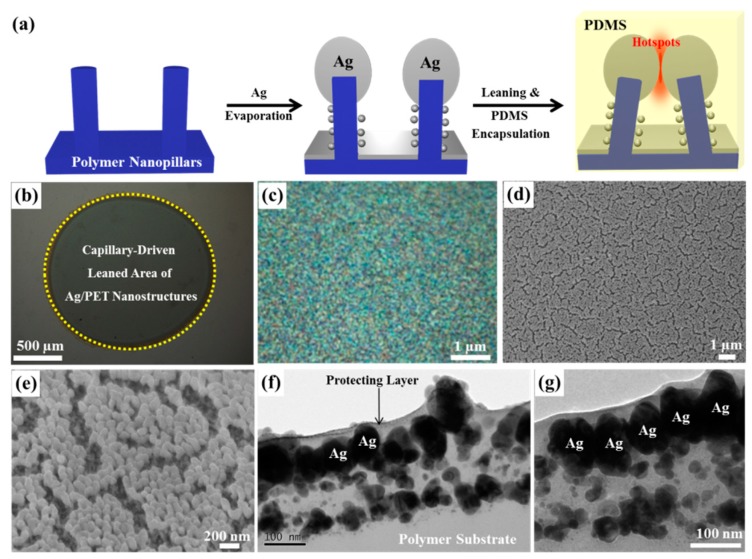
Fabrication of molecule-sensitive and selective nanogap-enhanced surface-enhanced Raman spectroscopy (SERS) sensing platform. (**a**) Schematic illustration of the fabrication process of Ag/PET nanopillars. (**b**) Optical image of the area within which the Ag/PET nanopillars leaned under surface tension (inside the dotted circle), and the as-prepared upright Ag/PET nanostructures (outside the dotted circle). (**c**) Dark field image of the leaning Ag/PET nanopillars, showing strong Rayleigh scattering from the Ag NPs. Scanning electron microscopy (SEM) images of the leaning Ag/PET nanopillars: (**d**) top view, and (**e**) tilted view. (**f**,**g**) Cross-sectional transmission electron microscopy (TEM) images of the collapsed Ag/PET nanopillars.

**Figure 2 nanomaterials-09-00619-f002:**
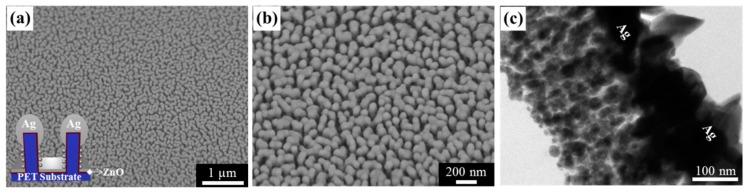
Fabrication of mechanically reinforced Ag nanostructures deposited onto soft polymer nanopillars. (**a**,**b**) SEM images of Ag nanostructures deposited onto ZnO/PET nanopillars. The inset shows a schematic cross-sectional structure of the mechanically reinforced Ag nanostructures. An 8.4-nm-thick ZnO thin film was conformally deposited by atomic layer deposition onto the plasma-treated PET nanopillars. The PET nanopillars had the same aspect ratios as the nanopillars used to assemble the capillary-induced leaning structures. (**c**) Cross-sectional TEM images of the Ag nanostructures after applying DI water. The mechanically reinforced Ag nanostructures stood upright, even after applying capillary forces, due to the deposition of a thin rigid ZnO layer. The TEM image suggests that Ag was grown under a wetting mode on the ZnO surface.

**Figure 3 nanomaterials-09-00619-f003:**
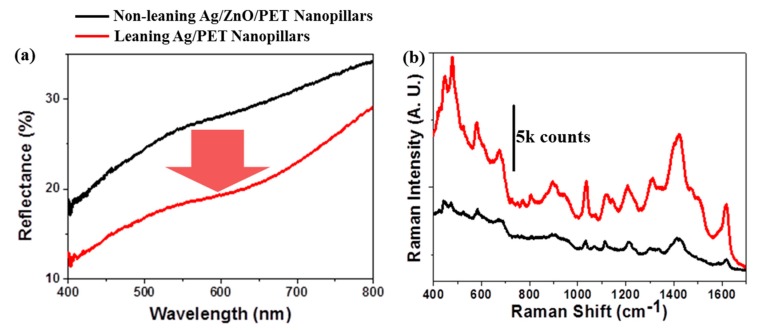
Optical and SERS properties of the non-leaning and leaning Ag nanostructures. (**a**) Reflectance spectra measured from the non-leaning ZnO-deposited Ag nanopillars and leaning Ag nanostructures. The reflectance intensity over the entire visible range decreased within the area containing the capillary-induced leaning Ag NPs due to the enhanced near-field coupling effects. (**b**) Raman spectra of methylene blue (MB) absorbed onto non-leaning and leaning Ag NPs. The excitation laser wavelength was 632.8 nm.

**Figure 4 nanomaterials-09-00619-f004:**
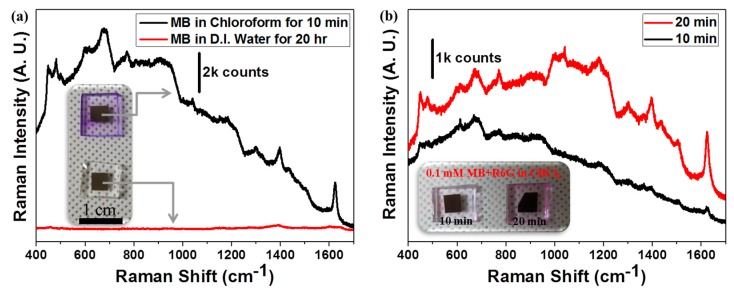
Nanofiltration-membrane-encapsulated SERS substrates that permitted the selective permeation of molecules. (**a**) Molecules infiltrated the encapsulated SERS substrate based on the degree of PDMS swelling in different solvents. PDMS swelled in contact with nonpolar chloroform, whereas it did not swell in polar water. Therefore, methylene blue (MB) molecules dissolved in chloroform could diffuse into the PDMS polymer network and toward hotspot regions of the SERS substrate. As time passed, the PDMS-encapsulated device soaked in chloroform turned blue due to MB infusion, whereas the PDMS-encapsulated device soaked in water maintained its transparency, even after a 20 h incubation period (inset). (**b**) Selective permeation based on molecular size. The encapsulated SERS substrate was soaked in MB- or rhodamine 6G (R6G)-containing chloroform solutions. Only the MB SERS spectrum was observed. The small MB (MW = 319.85 g/mol) molecules alone could diffuse into the PDMS membrane, and the large rhodamine 6G (MW = 497.02 g/mol) molecules were excluded.

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
