# Peer review of "Highly Sensitive and Selective Nanogap-Enhanced SERS Sensing Platform"

_nanomaterials, 2019, doi:10.3390/nano9040619_

Reviewer 1 Report

The authors report a new method for producing a sensitive SERS substrate through the leaning of metallized PET nanopillars produced by plasma etching. Moreover, the authors demonstrate an idea of achieving selective detection of target analytes through size exclusion by employing encapsulation of their SERS substrates into a PDMS membrane. The manuscript is well written and the results support the conclusions made by the authors. Moreover, I find the idea described herein novel enough to warrant publication in a prestigious journal such as Nanomaterials. Before I recommend acceptance, however, I would like to see the authors address the following concerns:

1. It is not very clear in the introduction whether the authors are the first ever researchers to apply dry etching of PET for generating nanopillars. References are given in the text regarding metallic nanopillars and polymer etching in general. If others have etched polymers before for the same purpose (plasmonic structures) their work will have to be mentioned. In this case, the importance and novelty of the authors’ work must be clearly spelled out within that frame of reference.  If, on the other hand, the authors are the first to document these findings, then they should make this achievement more prominent in the text.

2. To me, the manuscript demonstrates 2 different concepts that are not necessarily interdependent. For example, the method of selective detection described in the text (use of a PDMS membrane) is not exclusive to this type of SERS sensors, but could be used with any other SERS substrate. Although I do not disagree that the two methods can be presented together, as the authors do, I would wish for a more thorough presentation of the work that led to the development of the SERS substrate, assuming that this is the main novelty presented here. For example, are the reported etching conditions optimal or not? Perhaps the authors would like to discuss their experimental observations on how plasma conditions affect product characteristics (morphology, SERS enhancement)? Results and recommendations on how the etching environment can be improved towards achieving higher pillar density/SERS EF would be enlightening to the reader.

3. One final comment: “Areal density” is not a very well established term and is frequently used to described something else (data storage capacity in a hard disk). I would like to recommend to the authors to consider an alternative, more established term, such as “surface density”, or “area density”

Author Response

Comment: The authors report a new method for producing a sensitive SERS substrate through the leaning of metallized PET nanopillars produced by plasma etching. Moreover, the authors demonstrate an idea of achieving selective detection of target analytes through size exclusion by employing encapsulation of their SERS substrates into a PDMS membrane. The manuscript is well written and the results support the conclusions made by the authors. Moreover, I find the idea described herein novel enough to warrant publication in a prestigious journal such as Nanomaterials. Before I recommend acceptance, however, I would like to see the authors address the following concerns:

1. It is not very clear in the introduction whether the authors are the first ever researchers to apply dry etching of PET for generating nanopillars. References are given in the text regarding metallic nanopillars and polymer etching in general. If others have etched polymers before for the same purpose (plasmonic structures) their work will have to be mentioned. In this case, the importance and novelty of the authors’ work must be clearly spelled out within that frame of reference.  If, on the other hand, the authors are the first to document these findings, then they should make this achievement more prominent in the text.

Response: We appreciated your kind comments. To the best of our knowledge, our methods for producing SERS substrates through the leaning of metallized PET nanopillars produced by plasma etching have been already published two years ago (Adv. Funct. Mater. 2017, 27, 1703376.). That is the reason why we did not mention the importance and novelty of the fabrication method in this manuscript.

2. To me, the manuscript demonstrates 2 different concepts that are not necessarily interdependent. For example, the method of selective detection described in the text (use of a PDMS membrane) is not exclusive to this type of SERS sensors, but could be used with any other SERS substrate. Although I do not disagree that the two methods can be presented together, as the authors do, I would wish for a more thorough presentation of the work that led to the development of the SERS substrate, assuming that this is the main novelty presented here. For example, are the reported etching conditions optimal or not? Perhaps the authors would like to discuss their experimental observations on how plasma conditions affect product characteristics (morphology, SERS enhancement)? Results and recommendations on how the etching environment can be improved towards achieving higher pillar density/SERS EF would be enlightening to the reader.

Response: Thank you very much for your great idea. As you commented, improving selectivity is a very important issue in the SERS community. Therefore, you may think that we demonstrate two different concepts. Now, we are performing optimization of SERS substrates depending on process parameters, and thereby nanopillar morphologies and SERS EF. Since there are many operational parameters, it still needs some time to have optimized process conditions. I hope you could be a reviewer for our next paper.

3. One final comment: “Areal density” is not a very well established term and is frequently used to described something else (data storage capacity in a hard disk). I would like to recommend to the authors to consider an alternative, more established term, such as “surface density”, or “area density”

Response: As you recommended, I used “area density” in the revised manuscript. Overall, I would like to give you my great gratitude.

Reviewer 2 Report

The manuscript describes the fabrication of a SERS substrate composed of leaning PET/Ag nanoposts that gives rise to a high areal distribution of hotspots and a SERS enhancement factor on the order of 10^7.  The work is of potential interest to Nanomaterials readers, but the following suggestions/comments/questions should be addressed:

1) The figure depicting the structure being studied is introduced in the Results and Discussion section, long after the structure is introduced in both the Introduction and Methods sections. I recommend putting a reference to Figure 1a) when the structure is first introduced. 

2) The cartoon depicting the PET/Ag/ZnO structure in Figure 2 should be improved. It is impossible to discern what is going on in that figure.   I suggest making a larger Figure 2a cartoon similar to that in Figure 1a, and moving the SEM and TEM images to Figure 2b - 2d, similar to the layout in Figure 1 a - d.

3) Figures 1f and g are to be cross-sectional TEM images of the leaning nanoposts.  Is the cross-section in xy plane or the yz plane, with the z-axis being in the direction of the post axis?  That is, are we looking down on the posts or are we looking at them from the side?  It is difficult to see the 'leaning posts' described.   

4) Perhaps I am missing the initial definition in the text, but the only reference I saw to "MB" was in the caption to Figure 3.  If this is truly the case, please define within the text.  

5)The authors allude to the MB molecule being able to penetrate the PDMS, which 'swells in the presence of PDMS'... However, the SERS studies corresponding to Figure 3 are for aqueous MB.  Is the same mechanism allowing the MB to reach the hotspots when water is the solvent?  Is this expected to be the case for more realistic analytes of interest?   

6) Were the authors limited to 632 nm for the excitation wavelength?  Any indication of performance at 532nm?

Author Response

Comment: The manuscript describes the fabrication of a SERS substrate composed of leaning PET/Ag nanoposts that gives rise to a high areal distribution of hotspots and a SERS enhancement factor on the order of 107.  The work is of potential interest to Nanomaterials readers, but the following suggestions/comments/questions should be addressed:

Comments 1: The figure depicting the structure being studied is introduced in the Results and Discussion section, long after the structure is introduced in both the Introduction and Methods sections. I recommend putting a reference to Figure 1a) when the structure is first introduced.

Response 1: We appreciated your valuable comments. As you suggested, we put a reference to Figure 1a when the structure is first introduced.

Comments 2: The cartoon depicting the PET/Ag/ZnO structure in Figure 2 should be improved. It is impossible to discern what is going on in that figure.   I suggest making a larger Figure 2a cartoon similar to that in Figure 1a, and moving the SEM and TEM images to Figure 2b - 2d, similar to the layout in Figure 1 a - d.

Response 2: As you recommended, we modified Figure 2 in the revised manuscript. Please see the revised manuscript.

Comments 3: Figures 1f and g are to be cross-sectional TEM images of the leaning nanoposts.  Is the cross-section in xy plane or the yz plane, with the z-axis being in the direction of the post axis?  That is, are we looking down on the posts or are we looking at them from the side?  It is difficult to see the 'leaning posts' described.

Response 3: We are looking down on the post in the cross-sectional TEM images. It is difficult to see the ‘leaning PET nanoposts’, since PET nanoposts and protection layer have same contrast in TEM. Therefore, through cross-sectional TEM images, we can only see the clustered Ag NPs which are deposited on the top of PET nanoposts.

Comments 4: Perhaps I am missing the initial definition in the text, but the only reference I saw to "MB" was in the caption to Figure 3.  If this is truly the case, please define within the text.

Response 4: We already put the initial definition of MB at Line 77 of the original manuscript. I appreciated your valuable comments.

Comments 5: The authors allude to the MB molecule being able to penetrate the PDMS, which 'swells in the presence of PDMS'... However, the SERS studies corresponding to Figure 3 are for aqueous MB.  Is the same mechanism allowing the MB to reach the hotspots when water is the solvent?  Is this expected to be the case for more realistic analytes of interest?

Response 5: In order to clearly state our mechanism, we modified texts in the revised manuscript. Changes are as followed:

The PDMS-encapsulated SERS substrates turned bluish after a 10 min incubation period in a 1 mM MB chloroform solution (inset of Figure 4a). The SERS spectrum clearly shows that the MB SERS signal (black line in Figure 4a) was visible in the SERS substrates dipped for 10 min in the chloroform solution. However, the color of the PDMS-encapsulated SERS substrates dipped in the 1 mM MB aqueous solution did not change, even after a 20 hr incubation period. In this case, only a PDMS background SERS signal was measured from the SERS substrates dipped for 20 hr in the aqueous solution (Figure S6) [37].

Regarding on the more realistic analytes of interest, we plan to detect pesticides using our SERS platform in the near future. We appreciated your comments.

Comments 5: Were the authors limited to 632 nm for the excitation wavelength? Any indication of performance at 532nm?

Response 5: As we previously reported, Ag nanostructures also showed a very good SERS performance at 532 nm laser excitation (RSC Adv., 2018, 8, 6444–6451). Please check our previous literature.

Reviewer 3 Report

 Authors describe a new method for generating high areal density plasmonic nanogaps through the direct maskless dry etching of soft polymer surfaces combined with capillary leaning effects. The introduction and result section looks good and experimental demonstration excellent. Can the authors explain what do they mean by the enhancement factor number of 1.5 × 107?  

Overall, the paper is written relatively well. However, the presentation style could still be improved. For example, long paragraphs could be broken into short and clear paragraphs.

The method section only describes in words.  I would like to see the schematics, showing details such as optical signal and the direction in which these signals were fed. Show a schematic of experimental set up prepared if any, to study plasmonic activity. 

The manuscript does not have a single table listing the important material parameters or results. I encourage authors to create a table and present the sensitivity information available in the published papers and compare those with the authors. The conclusion section should include at least a few important quantitative results derived from this work. 

Author Response

Comment: Authors describe a new method for generating high areal density plasmonic nanogaps through the direct maskless dry etching of soft polymer surfaces combined with capillary leaning effects. The introduction and result section looks good and experimental demonstration excellent. Can the authors explain what do they mean by the enhancement factor number of 1.5 × 107

Response: As we already stated in the original manuscript, the SERS EF was considered to be the main parameter that determined the performance of the SERS substrate itself. SERS EF is how much the Raman signal is amplified with respect to normal conditions. Please see the Figure S4 for the calculation of average SERS EF of our SERS substrates.

Comment: Overall, the paper is written relatively well. However, the presentation style could still be improved. For example, long paragraphs could be broken into short and clear paragraphs.

The method section only describes in words.  I would like to see the schematics, showing details such as optical signal and the direction in which these signals were fed. Show a schematic of experimental set up prepared if any, to study plasmonic activity.

Response: As you recommended, we broke long paragraphs into short and clear paragraphs. We appreciated your valuable comments. Since we used conventional optical setup, we did not show a schematic of experimental setup. Please check the literature on the optical setup (Opt. Express, 2018, 26, 6439).

Comment: The manuscript does not have a single table listing the important material parameters or results. I encourage authors to create a table and present the sensitivity information available in the published papers and compare those with the authors. The conclusion section should include at least a few important quantitative results derived from this work.

Response: There are countless papers on the SERS substrates and their applications. Therefore we strongly believe that creation of a table on sensitivity of SERS substrates is impossible.

Round  2

Reviewer 1 Report

I would like to thank the authors for their responses and for referring me to their previously published work (Park, S.-G, Adv. Funct. Mater. 2017, 27, 1703376). Reading that article, I realized that there is a big overlap with the present manuscript. Therefore, I would like to ask the authors to present more clearly in their introduction the specific aspects of their method that are new in this manuscript, also referring clearly to the part of this work that has been reported earlier. The statement (lines 56-57) ”Here, we report a new method for generating high area density plasmonic nanogaps through the direct maskless dry etching of soft polymer surfaces combined with capillary leaning effects” is potentially misleading to the reader who is not familiar with the authors’ previously published work. Although the authors refer to their previous work in the text (line 36), a direct connection between this work and prior art is not established.

Finally, if this method is for building nanomaterials is not as novel as originally presented, this begs the question: what is really novel that makes this piece of work worth-reporting in such a prestigious journal? I expect the authors to make that very clear in the text. I am sorry for being so peaky, but when the Conclusions section begins with (lines 249-251) “We reported a method for generating high areal density plasmonic hotspots through the direct maskless dry etching of soft polymer surfaces combined with capillary leaning effects. The capillary-driven leaning effects led to the assembly of high areal density plasmonic nanogaps of the Ag nanoparticles/PET nanopillars hybrid structures” the readers wonders how much of this is really new and not previously reported. Both the fabrication details and plasmonic activity of these substrates have been documented before [Ref. 17].

In light of the fact that the novelty of the SES substrate is not as high as it appears, the authors may wish to shift the emphasis of the manuscript to the second concept that they describe, namely the selective detection capabilities of their system. Although this is not unique to their substrates, this method is an interesting one and worth reporting.

Thank you for considering my comments.

Author Response

Comment 1: I would like to thank the authors for their responses and for referring me to their previously published work (Park, S.-G, Adv. Funct. Mater. 2017, 27, 1703376). Reading that article, I realized that there is a big overlap with the present manuscript. Therefore, I would like to ask the authors to present more clearly in their introduction the specific aspects of their method that are new in this manuscript, also referring clearly to the part of this work that has been reported earlier. The statement (lines 56-57) ”Here, we report a new method for generating high area density plasmonic nanogaps through the direct maskless dry etching of soft polymer surfaces combined with capillary leaning effects” is potentially misleading to the reader who is not familiar with the authors’ previously published work. Although the authors refer to their previous work in the text (line 36), a direct connection between this work and prior art is not established.

Response 1: We appreciated your valuable comments. As you suggested, we put an emphasis the novelty of the present work, which is the selective detection capabilities of our plasmonic system. Please check the revised manuscript.

Comment 2: Finally, if this method is for building nanomaterials is not as novel as originally presented, this begs the question: what is really novel that makes this piece of work worth-reporting in such a prestigious journal? I expect the authors to make that very clear in the text. I am sorry for being so peaky, but when the Conclusions section begins with (lines 249-251) “We reported a method for generating high areal density plasmonic hotspots through the direct maskless dry etching of soft polymer surfaces combined with capillary leaning effects. The capillary-driven leaning effects led to the assembly of high areal density plasmonic nanogaps of the Ag nanoparticles/PET nanopillars hybrid structures” the readers wonders how much of this is really new and not previously reported. Both the fabrication details and plasmonic activity of these substrates have been documented before [Ref. 17].

Response 2: As you recommended, we put a clear novelty on highly selective SERS substrate of the current works. We have modified Conclusions putting an emphasis of the enhanced selectivity in the revised manuscript. Please see the revised manuscript.

Comment 3: In light of the fact that the novelty of the SES substrate is not as high as it appears, the authors may wish to shift the emphasis of the manuscript to the second concept that they describe, namely the selective detection capabilities of their system. Although this is not unique to their substrates, this method is an interesting one and worth reporting.

Response 3: Once again, we appreciated your valuable comments. As you suggested, we put the emphasis of the manuscript on the selective detection capabilities of our system. As you mentioned, since our encapsulation method can be applicable to general SERS substrate, we strongly believe that our method should appeal to the broad readership of the journal. Thank you very much for your valuable comments.

Round  3

Reviewer 1 Report

I would like to thank the authors making appropriate changes to the script.

Line 60 (end of sentence), the authors must cite Ref 17 (previous work, where this method was first reported).

Author Response

Thank you for your kind comments. As you commented, we have cited Ref. 17 at the Line 60.

We also put an emphasis the novelty of the present work more clearly, which is the selective detection capabilities of our plasmonic system. Please check the revised manuscript. Once again, thank you very much for your valuable comments.